# Loco-Regional Control and Sustained Difference in Serum Immune Protein Expression in Patients Treated for p16-Positive and p16-Negative Head and Neck Squamous Cell Carcinoma

**DOI:** 10.3390/ijms24043838

**Published:** 2023-02-14

**Authors:** Karl Sandström, Ylva Tiblom Ehrsson, Felix Sellberg, Hemming Johansson, Göran Laurell

**Affiliations:** 1Department of Surgical Sciences, Uppsala University, 751 85 Uppsala, Sweden; 2Department of Immunology, Genetics and Pathology, Uppsala University, 751 85 Uppsala, Sweden; 3Department of Oncology-Pathology, Karolinska University Hospital, 171 76 Stockholm, Sweden

**Keywords:** P16, serum, cytokines, protein expression, head and neck cancer

## Abstract

The main prognostic factors for patients with head and neck cancer are the tumour site and stage, yet immunological and metabolic factors are certainly important, although knowledge is still limited. Expression of the biomarker p16INK4a (p16) in oropharyngeal cancer tumour tissue is one of the few biomarkers for the diagnosis and prognosis of head and neck cancer. The association between p16 expression in the tumour and the systemic immune response in the blood compartment has not been established. This study aimed to assess whether there is a difference in serum immune protein expression profiles between patients with p16+ and p16- head and squamous cell carcinoma (HNCC). The serum immune protein expression profiles, using the Olink^®^ immunoassay, of 132 patients with p16+ and p16- tumours were compared before treatment and one year after treatment. A significant difference in the serum immune protein expression profile was observed both before and one year after treatment. In the p16- group, a low expression of four proteins: IL12RB1, CD28, CCL3, and GZMA before treatment conferred a higher rate of failure. Based on the sustained difference between serum immune proteins, we hypothesise that the immunological system is still adapted to the tumour p16 status one year after tumour eradication or that a fundamental difference exists in the immunological system between patients with p16+ and p16- tumours.

## 1. Introduction

Malignant tumours of the head and neck is the sixth-most prevalent cancer type in the world, and its prevalence has risen, primarily because of an increase in oropharyngeal cancers that are *human papilloma virus* (HPV)-positive [1]. More than 90% are squamous carcinoma (HNSCC) arising in the mucosal lining of the upper aerodigestive tract. A distinctive feature of HNSCC is its anatomical site-specific pattern, which has implications for treatment, recurrence, and survival. More than half of the patients experience loco-regional or distance relapse despite aggressive multimodal treatment. 

The most widely accepted and used system for staging HNSCC is The American Joint Committee on Cancer/International Union Against Cancer TNM (TNM) staging system. The staging system is mainly anatomical, taking into account the size of the tumour and its relationship to the surrounding tissue, its spread to lymph nodes, or metastatic spread. The system is relatively simple and user-friendly, with high acceptance and compliance worldwide, but the hazard discrimination within each stage could be improved. For more precise patient outcome prediction tumour-related factors (i.e., biomarkers) such as genomic alternations, altered signalling pathways and factors in the tumour microenvironment need to be included [2]. 

The latest version of the TNM system (8th), published in 2017, incorporates one biomarker for head and neck cancer, p16INK4a (p16) [3]. P16 is as a surrogate marker for high-risk HPV infection in oropharyngeal carcinomas (OPSCC) [4]. P16+ OPSSC is a distinct subtype, and patients with this subtype have a better prognosis than patients with p16- OPSCC [5,6]. Immunohistochemistry for p16 is available in most pathology laboratories and is relatively simple to perform with high sensitivity and specificity.

Immunological and metabolic factors undoubtedly play an important role in HNSCC prognosis. HPV-positive tumours are reported to have a higher infiltration of T cells than HPV-negative tumours [7]. The association between p16 expression in the tumour and the systemic immune response in the blood compartment has not been established.

The aim of the present study was to analyse whether there is a difference in serum immune protein expression profiles between patients with p16+ and p16- HNSCC pre-and one-year post-treatment. 

## 2. Results

### 2.1. Patient and Tumour Characteristics at Baseline

The p16+ group had a lower stage, were younger, received chemoradiotherapy more often, and had a better prognosis compared to the p16- group. The patient characteristics are provided in Table 1.

### 2.2. Serum Immune Proteins Expression at Baseline

Statistically significant mean differences in protein expression between the p16+ group and p16- group were found for 5 of 92 proteins (PTN, ARG1, CD28, TNFRSF12A, and CD70); with lower expression in the p16+ group (Figure 1).

### 2.3. Serum Immune Protein Expression at Baseline and Effect on Cumulative Incidence of Failure

A significant effect on the cumulative incidence of failure was independently observed for the four proteins at baseline. Patients with high expression of CD244, TNFRSF21, IL12, or IL5 showed a lower cumulative incidence of failure during the two-year follow-up period, as shown in Figure 2.

### 2.4. The Interaction between p16 and Serum Immune Protein Expression at Baseline in Relation to Treatment Failure 

A significant interaction was observed between p16 and four proteins: IL12RB1, CD28, CCL3, and GZMA. The NPX values for each protein were grouped into low and high groups, using the median as the cut-off. Low expression of each protein and p16- conferred a higher failure rate; however, low expression and p16+ did not affect the outcome (Figure 3). The strongest interaction was observed between p16 and a low expression of IL12RB1.

### 2.5. Serum Protein Expression 12 Months after Primary Treatment in Patients without Failure

Two years after the primary treatment, 92 patients had no treatment failure, and the patient characteristics are presented in Table 2. Of the 40 patients with treatment failure, 16 had a recurrence but were still alive at 24 months. The tumour sites of patients with treatment failure are presented in Table 3. The serum immune protein levels at 12 months in patients without treatment failure differed significantly for twenty-two proteins, with lower expression in the p16+ tumour group (Figure 4). All but one (ARG1) of the proteins with significantly lower expression at baseline in the p16+ tumour group also showed lower expression at 12 months—that is, PTN, TNFRSF12A, CD28, and CD70.

## 3. Discussion

In this prospective exploratory study of a mixed cohort of head and neck cancer patients, we compared the expression of serum immune proteins in two groups of patients: those with p16+ HNSCC and those with p16- HNSCC. Different expressions of serum immune proteins were seen before the initiation of treatment, as well as in blood serum samples taken at the one-year follow-up appointment. These results indicate that there might be a difference in host immunity between patients with p16+ and p16- HNSCC unrelated to the presence of a viable tumour. 

Patients with p16+ OPSSC generally have a better prognosis and outcome compared to p16- OPSCC [5,6]. The majority of HPV-negative HNSCC are p16-, mainly due to the loss of chromosome 9p, including CDKN2A (9p21). The loss of chromosome 9p, together with mutations in TP53 and loss of chromosome 3, are the most frequent genetic alterations in the early carcinogenesis of HNSCC [8,9]. In OPSSC, p16+ is strongly correlated with the expression of HPV E7 oncoprotein [9] and is a surrogate marker for a transcriptionally active infection with a high-risk human papillomavirus (HPV), most commonly HPV-16 [10]. 

In the p16+ group, 70 of the 82 patients had OPSSC. The overexpression of p16 is known to occur in a few HPV-negative tumours in the oral cavity, larynx, hypopharynx, nasopharynx, and nasal sinuses [11]. The reason for the overexpression of p16 in HPV-negative tumours is not well understood. The overexpression could be attributed to non-functional missense mutations with inactive p16 or the expression of high levels of active p16 due to mutations in the histone methyltransferase NSD1 [11]. The impact of p16 overexpression on outcomes in non-OPSCC patients remains uncertain, although some studies have suggested an improved outcome [12,13,14,15]. 

The identification of immune proteins and the association between protein expression and prognosis in HPV-induced cancer has been widely studied in experimental and clinical research. However, there are only a few comprehensive studies on the serum immune profiles of HNSCC patients. The approach undertaken in the present study enabled the study of systemic serum immune proteins in patients with p16+ and p16- HNSCC tumours, which were characterised by blood samples taken before treatment and one year after the termination of treatment. 

Analysing the entire cohort at baseline, when all patients were affected by a HNSCC, 5 out of 92 proteins had significantly lower means in the p16+ group: PTN, ARG1, CD28, CD70, and TNFRSF 12A. At one year, in patients without treatment failure, the immune profiles demonstrated a clear-cut distinction, as the immune profiles of 22 proteins differed between the two groups, all with lower means in the p16+ group. All but one protein (ARG1) that differed at baseline also displayed lower means after one year. A few earlier studies have assessed the clinical significance of the serum levels of these four proteins in patients with malignant diseases. 

CD28 is an important immune checkpoint protein on the surface of CD8+ T cells. High serum levels of CD28, among other immune checkpoint proteins, have been associated with worse prognosis in prostate cancer [16]. 

CD70 is a cytokine belonging to the tumour necrosis factor superfamily and is expressed only on activated T and B cells, dendritic cells, and NK cells. The interaction of CD70 with its receptor, CD27, results in B- and T-cell activation. In a recent study, Swiderska et al. found evidence of a correlation between the serum CD70 concentration and overall survival in patients with ovarian cancer [17]. The authors also showed that higher concentrations of CD70 in the peritoneal fluid were associated with peritoneal carcinomatosis. Elevated expression in ovarian cancer has been associated with resistance to cisplatin therapy [18]. 

TNFRSF12A (FN14, TWEAKR, and CD266—tumour necrosis factor receptor superfamily 12A) is the receptor for TWEAK, with low expression in most tissues. TNFRSF12A appears to be an important protein in the proliferation, invasion, and migration of tumour cells and is overexpressed in many different cancers [19,20]. In a case–control study, Chang et al. identified that the serum levels of TNFRSF12A were lower in patients with non-small-cell lung cancer than in healthy controls and were not correlated with cell type, TNM stage, or metastasis status [21]. 

PTN (pleiotrophin) is a secreted multifunctional protein that affects cell growth and survival, migration, angiogenesis, and tumorigenesis. Increased serum levels have been associated with metastatic disease in patients with prostate cancer [22]. In patients with colorectal cancer, the serum levels were higher than those in the non-cancer control group [23]. In patients with small-cell lung cancer, the serum levels were elevated compared to those in patients with benign lung disease and were associated with a worse prognosis [24]. Moreover, in a study by Souttou et al., elevated levels of PTN normalised after tumour resection [25]. 

The main function of Arg-1 (Arginase 1) is converting L-arginine to urea and L-ornithine in the liver. Arg-1 is also expressed in bone marrow and some hemopoietic cells. M2 macrophages can express ARG-1 mediated by IL-4 and IL-13. Stroma cells and tumour cells may also express ARG-1. Increased levels have been reported in different cancers, but the effect on the prognosis is uncertain. High levels in the tumour microenvironment are most likely immunosuppressive. Increased Arg-1 activity is seen in many tumours, such as head and neck cancer, and increased plasma levels have also been observed in ovarian cancer patients [26,27]. 

The difference in the serum protein expression at diagnosis between patients with p16+ and p16- draws our attention to immunological effects induced by a history of persistent high-risk HPV infections in many patients with p16+ tumours. Most HPV infections are cleared by the immune system; however, high-risk HPV can evade immune surveillance. HPV-encoded E6 and E7 oncoproteins have been shown to play an essential role in HPV escape from the immune system by suppressing NF-ĸB activity [28,29]. The immunological effects of high-risk HPV infections are complex and involve a wide range of mechanisms within the immune system, including CD8 cell effects in HPV-positive HNSCC [30]. The protein alterations driven by HPV-induced oncoproteins may contribute to the significantly lower expression levels of the five proteins found in the serum at diagnosis in patients treated for p16+ HNSCC. However, the most remarkable result to emerge from this study is the large difference in the serum immune protein profile at one year after termination of the treatment in the two groups of recurrence-free patients. One could assume that there would not be any difference in the serum immune protein expression profile if it depended only on the tumour–host interaction, which can most probably be seen prior to treatment. The exact role of this difference in long-term outcomes should be established in future studies. However, this finding allows us to speculate that an adaption of the immunological system to the tumour p16 status is either still maintained even one year after tumour eradication or that there is a fundamental difference in the patient’s immunological system, perhaps conveying a susceptibility to HPV infection from the start. Given that our findings are based on the p16 status and not the HPV status, the findings should be treated with considerable caution, and unintended bias cannot be ruled out.

The effect of the serum protein levels at baseline on failure was also analysed. In the whole group, low levels of CD244, TNFRSF21, IL12, and IL5 all independently conferred a higher cumulative incidence of failure. Low levels of CD244, the ligand of CD48, expressed in various immune cells such as NK cells [31] with dual immunomodulatory effects [32], had a strong effect on failure.

In the interaction analysis, low baseline levels of IL12RB1, CD28, CCL3, and GZMA conferred worse outcomes in the p16- group but not in the p16+ group. The strongest interaction was with IL12RB1, which forms a functional high-affinity receptor for IL12, in association with IL12RB2. 

Recognising the limited knowledge on which role these proteins play in the immune response and resistance to tumour recurrence in p16+ and p16- HNSCC, respectively, it is reasonable to consider how the levels of immune proteins in the blood represent events in the tumour tissue and the tumour microenvironment. Knowledge about the interplay between these compartments is still limited and has a strong reason to be explored in detail, particularly for the development of biomarkers of prognostic significance in the HNSCC environment [33,34,35,36].

Even though there are inherent limitations in our approach of mapping the immune profiles in p16+ HNSSC and p16- HNSCC, the results may add some critical information to better understand the background of the differences in treatment outcomes seen in the two groups. Advances in this field can also be valuable when seeking effective individualised treatments based on the HPV status. In the analysis of tumour-promoting biological processes in HNSCC, immune profiling is the most valuable for identifying either an inflammatory or immunosuppressive phenotype. Tumours are prone to infiltration by various immune cells. In contrast to analysis of the blood compartment, immune cell profiling of tumour tissues can identify the immune signature and differentiate between hot, cold, and excluded tumours [37,38]

In a recent review, Bhat et al. summarised the present knowledge on the contributing factors of secreted proteins in the TME for the importance of the aggressive nature of HNSSC, with special reference to the HPV status [39]. Repeated blood assessments may have suitable properties for follow-up patients previously treated for HNSSC. The expression of immune proteins has recently been characterised in tumour samples from patients with HPV-positive and HPV-negative OPSSC; Ramqvist et al. analysed samples from 59 patients using the same technique as in our study (Olink, PEA) but also used two other panels in addition to the IMMUNO-ONCOLOGY panel. In tumour tissues, significant differences in the expression of 34 proteins were observed in HPV-positive and HPV-negative tumours [40]. Although the study analysed a different compartment (tumour tissue), some comparisons with our study are still interesting. A discrete association with the findings in the present study can be noticed; TNFRSF12A showed lower expression in HPV-positive tumours, in line with our results from the serum analysis. 

To our knowledge, no earlier published studies have compared how immune protein expression in the serum is affected by the p16 status. A difference in peripheral cytokine levels in HPV+ and HPV- patients has been reported previously; however, with the use of different techniques, it is difficult to draw direct comparisons. Mytilineos et al. measured cytokines in the blood before and up to one year after the treatment of patients with HNSCC. The Granzyme B (GZMB) and MIP-1β (CCL4) levels were increased during the treatment course in the HPV-positive group, but there were no differences before or 12 months after treatment [41]. Dickinson et al. aimed to identify cost-effective serum markers to improve decisions at risk of misclassification based on the p16 status alone. Of the 174 analysed serum proteins before treatment, three differed between HPV-positive and HPV-negative patients: complement component C7 (C7), apolipoprotein F (ApoF), and galectin-3-Binding Protein (LGALS3BP) [42]. Aarstad et al. reported no differences depending on the HPV status in an acute-phase cytokine profile before treatment in 144 patients with HNSCC [43].

This study has some limitations, and information on the HPV status could further illustrate possible differences in the immune response. There is some heterogeneity regarding tumour sites, with most p16+ tumours located in the oropharynx, mirroring the clinical situation. In addition, the majority of p16+ tumours were stages I and II (TNM staging system, 8th version) compared to a more even distribution in the p16- group, which could affect the failure rate. Only blood samples were available for this study, which roughly mirrored the immune response in the TME. To further understand the interplay between tumour cells, simultaneous assessments of immune cells in the TME and the serum immune system are needed.

## 4. Materials and Methods

### 4.1. Study Design

The present study was part of a large prospective observational study performed at three head and neck cancer centres in Sweden. The inclusion criteria were curable, newly diagnosed untreated HNC and a performance status of 0–2, according to the WHO/ECOG Performance Status Scale. Exclusion criteria included previous treatment for malignant neoplasms within the last 5 years (except for skin cancer), immune therapy, severe alcoholism, cognitive impairments, or other inability to participate and understand Swedish. Blood was drawn from the patients before treatment (baseline) and one year after treatment termination. The samples were stored in serum at −70 °C at the Uppsala Biobank (approved RCC 2015-0025). This study was registered at ClinicalTrials.gov (NCT03343236). The present study is an extension of our earlier study aimed at determining the immune reaction to different treatment modalities [44].

The cohort in the present study consisted of 132 patients with a known p16 status and at least two years of follow-up; 82 patients were p16+, and 50 patients were p16-. All patients were staged according to the Union for International Cancer Control TNM 8 staging system and treated according to the Swedish National Guidelines for Head and Neck Cancer. Patients were treated at one of the participating university hospitals: Uppsala, Örebro, or Umeå. The patients were under surveillance with clinical examinations every three months, and radiological investigations were performed if recurrence was suspected. An evaluation of the treatment response in patients with oropharyngeal squamous cell carcinoma was performed using FDG-PET/CT 3 months after treatment termination. A representative study met with the included patients before treatment at one and two years after the termination of treatment. All patients were treated with curative intent and showed no evidence of generalised disease at enrolment in the study.

Treatment failure was defined as death due to any cause or recurrence within 24 months of the primary treatment. Disease recurrence was defined as viable cancer in a biopsy and/or salvage operation specimens after the primary treatment. For oropharyngeal cancer, neck dissection with viable cancer after primary radiotherapy or chemoradiation was defined as salvage surgery.

Blood was drawn before treatment (baseline) and 12 months after therapy, and the serum was analysed by Olink Proteomics LC using the Olink^®^ IMMUNO-ONCOLOGY immunoassay (Olink Proteomics, Uppsala, Sweden). The Olink^®^ IMMUNO-ONCOLOGY immunoassay measures 92 immuno-oncology-related human proteins using a proximity extension assay [45]. This panel offers the simultaneous analysis of protein biomarkers involved in adaptive immune responses, lymphocyte activation, inflammatory responses, and cytokine-mediated signalling pathways. The results are expressed as normalised protein expression (NPX), an arbitrary unit on the Log2 scale [46]. 

### 4.2. Statistical Analysis

Descriptive statistics presented distributions as the number of cases with percentages for categorical variables and as medians, together with their minimum and maximum values. Distributional differences in the categorical variables and p16 status were tested using Fisher’s exact test, and differences in the continuous variables were tested using the Mann–Whitney test. Associations between 92 protein expression levels and the p16 status were estimated using linear regression. The results from these models were presented as the mean difference (MD) (mean difference = protein log2 level of the p16+ group minus protein log2 level of the p16- group), 95% confidence interval (CI), and Wald *p*-value. Survival time was calculated from the date of treatment to the date of recurrence or death. For event-free patients, the time was calculated from the day of treatment to the date of the last clinical visit. Graphs illustrating the cumulative incidence of failure were calculated using the Kaplan–Meier method. The effect of the P16 status and serum immune protein levels (NPX values for each protein were grouped into low and high groups using the median as the cut-off.) on time to failure were estimated using proportional hazards regression and presented as hazard ratios (HR), together with 95% CIs and Wald *p*-values. Deviations from the proportional hazard assumptions were evaluated by testing the nonzero slope of the scaled Schoenfeld residuals over time. The reported *p*-values were two-sided, and the level of significance was set at 0.05.

## 5. Conclusions

In this study, we found significant differences in the serum immune protein expression profiles between patients with p16+ and p16- tumours. For four proteins (PTN, CD28, TNFRSF 12A, and CD70), differences in their expression were apparent both before treatment and one year after treatment in patients without treatment failure. In addition, a low expression of IL12RB1, CD28, CCL3, and GZMA before treatment in the p16- tumour group conferred a higher rate of failure. 

## Figures and Tables

**Figure 1 ijms-24-03838-f001:**
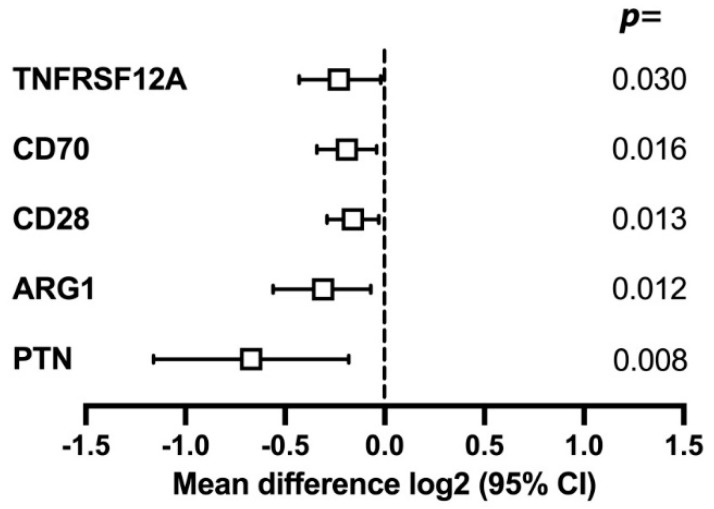
Forest plot on the effect of the p16 status on protein expression at baseline. Bars represent the confidence interval (CI), and the box represents the median.

**Figure 2 ijms-24-03838-f002:**
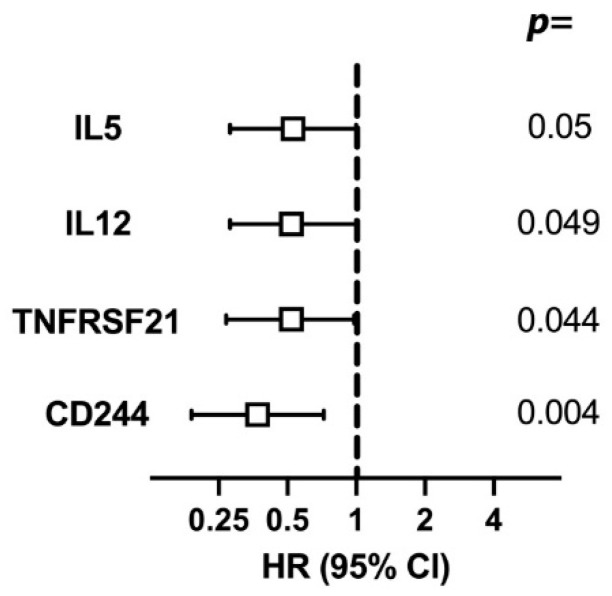
Univariable risk of treatment failure. Serum immune protein levels at baseline with an effect on the cumulative incidence of failure. Regression analysis. NPX values for each protein were grouped into a low and high group using the median as the cut-off. *X*-axis shows the odds ratio. Hazard ratio (HR). Bars represent the confidence interval (CI), and the box represents the median.

**Figure 3 ijms-24-03838-f003:**
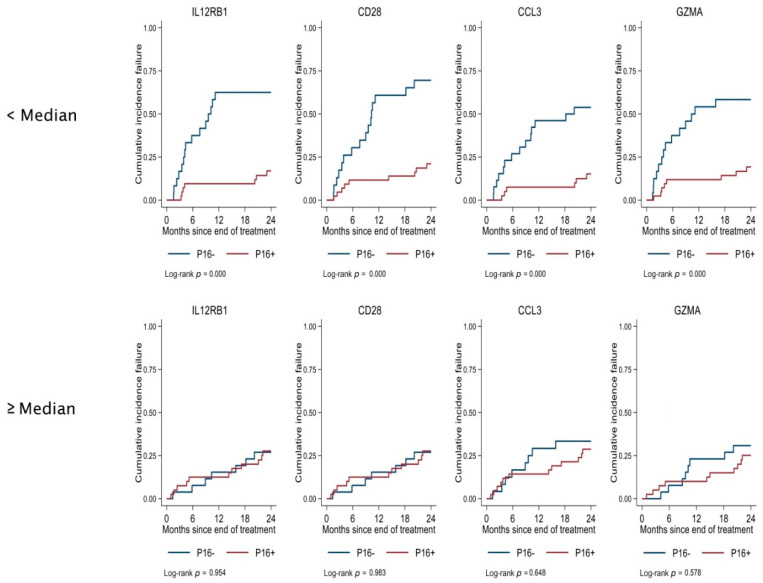
The interaction between p16 and proteins in relation to treatment failure. (˂ Median = Low values, ≥ High values. *Y*-axis: Cumulative incidence of failure. *X*-axis: Months since end of treatment).

**Figure 4 ijms-24-03838-f004:**
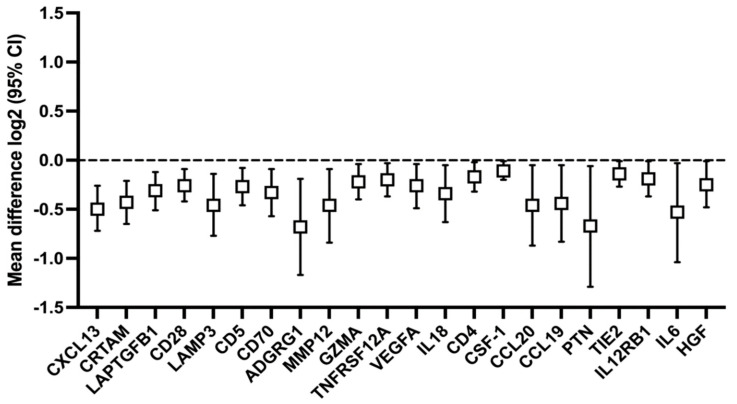
Forest plot on the effect of the p16 status on protein expressions at 12 months after the primary treatment in patients without failure within two years of the primary treatment. Only proteins with significant differences are shown, with increasing *p*-values from left to right (CXCL13 *p* = 0.000, HGF *p* = 0.042). Bars represent the confidence interval (CI), and the box represents the median.

**Table 1 ijms-24-03838-t001:** Characteristics of a cohort consisting of 132 patients with head and neck cancer. *p*-values for sex, site, stage, and treatment were calculated using Fischer’s exact test. For age, the Wilcoxon rank-sum test was used. (* Fischer’s exact test, ** Wilcoxon rank-sum test).

				**p16 Status**				
		**Total (N = 132)**		**p16+ (N = 82)**		**p16- (N = 50)**		***p*-Value**
**Age at Diagnosis. Mean (Median)**		62 (62)		60 (60)		65 (67)		0.003 *
		**Count**	**Column N %**	**Count**	**Column N %**	**Count**	**Column N %**	
**Sex**	Male	97	73%	64	78%	33	66%	0.16 **
	Female	35	27%	18	22%	17	34%	
**Site**	Oropharynx	75	57%	70	86%	5	10%	˂0.001 **
	Oral cavity	28	21%	6	7%	22	44%	
	Larynx	17	13%	1	1%	16	32%	
	Hypopharynx	5	4%	0	0%	5	10%	
	Nasopharynx	3	2%	1	1%	2	4%	
	Nose and nasal sinuses	3	2%	3	4%	0	0%	
	Unkown primary	1	1%	1	1%	0	0%	
**Stage**	I	61	46%	47	57%	14	28%	˂0.001 **
	II	26	20%	15	18%	11	22%	
	III	26	20%	18	22%	8	16%	
	IVa	16	12%	1	1%	15	30%	
	IVb	3	2%	1	1%	2	4%	
**Treatment**	Surgery as monotherapy	11	8%	4	5%	7	14%	˂0.001 **
	Radiotherapy as monotherapy	46	35%	33	40%	13	26%	
	Surgery and radiotherapy	23	17%	6	7%	17	34%	
	Chemoradiotherapy±surgery	41	31%	31	38%	10	20%	
	Radiotherapy+anti-EGFR±surgery	11	8%	8	10%	3	6%	

**Table 2 ijms-24-03838-t002:** Characteristics of patients without failure (recurrent tumour or death) within two years of the primary treatment.

				**p16 Status**				
		**Total (N = 92)**		**p16+ (N = 64)**		**p16- (N = 28)**		
**Age at Diagnosis. Mean (Median)**		61 (62)		59 (60)		65 (64)		
		**Count**	**Column N %**	**Count**	**Column N %**	**Count**	**Column N %**	**Chi Square**
**Sex**	Male	69	75%	50	78%	19	68%	0.3
	Female	23	25%	14	22%	9	32%	
**Site**	Oropharynx	55	60%	54	86%	1	4%	˂0.001
	Oral cavity	19	21%	5	8%	14	50%	
	Larynx	12	13%	1	2%	11	39%	
	Hypopharynx	1	1%	0	0%	1	4%	
	Nasopharynx	2	2%	1	2%	1	4%	
	Nose and nasal sinuses	2	2%	2	3%	0	0%	
	Unknown primary	1	1%	1	2%	0	0%	
**Stage**	I	52	57%	42	66%	10	36%	0.005
	II	19	21%	11	17%	8	29%	
	III	13	14%	9	14%	4	14%	
	IVa	7	8%	1	2%	6	21%	
	IVb	1	1%	1	2%	0	0%	
**Treatment**	Surgery as monotherapy	7	8%	3	5%	4	14%	0.004
	Radiotherapy as monotherapy	34	37%	26	41%	8	29%	
	Surgery and radiotherapy	13	14%	4	6%	9	32%	
	Chemoradiotherapy±surgery	30	33%	25	39%	5	18%	
	Radiotherapy+anti-EGFR±surgery	8	9%	6	9%	2	7%	

**Table 3 ijms-24-03838-t003:** p16 status and tumour site for patients with treatment failure (recurrent tumour or death) within two years of the primary therapy.

		Total		p16+		p16-	
		Dead (N = 24)	Recurrence (N = 16)	Dead (N = 5)	Recurrence (N = 13)	Dead (N = 19)	Recurrence (N = 3)
Site	Oropharynx	9	11	5	11	4	
	Oral cavity	7	2		1	7	1
	Larynx	4				4	1
	Hypopharynx	4				4	
	Nasopharynx		1				1
	Nose and nasal sinuses		1		1		

## Data Availability

Data are available upon request.

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
