# Peer review of "Loco-Regional Control and Sustained Difference in Serum Immune Protein Expression in Patients Treated for p16-Positive and p16-Negative Head and Neck Squamous Cell Carcinoma"

_ijms, 2023, doi:10.3390/ijms24043838_

Round 1

Reviewer 1 Report

the manuscript presented here attempts to relate entirely clinical aspects in head and neck tumours, a quite heterogeneous oncological disease entity. The authors divide the group of patients according to the presence or absence of the p16 marker, associating this with a profile of proteins involved in immune response. Although the article is well written and provides important clinical data, it seems to me that it should be transferred to a journal with a wider clinical audience, such as Current Oncology. While the article is interesting from a clinical perspective, it does not explain any molecular mechanisms or address a topic that is relevant to the journal's audience.

Minor,
Epidemiological data is needed in the introduction to understand the importance of head and neck cancer.

Author Response

Comment 1. The manuscript presented here attempts to relate entirely clinical aspects in head and neck tumours, a quite heterogeneous oncological disease entity. The authors divide the group of patients according to the presence or absence of the p16 marker, associating this with a profile of proteins involved in immune response. Although the article is well written and provides important clinical data, it seems to me that it should be transferred to a journal with a wider clinical audience, such as Current Oncology. While the article is interesting from a clinical perspective, it does not explain any molecular mechanisms or address a topic that is relevant to the journal's audience.

Response: We have been invited to contribute to the Special Issue “Proteomics and Metabolomics Approaches on Cancer Research”. In the present study we focus on assessing important differences in serum immune protein expression between patients with p16+ and patients with p16- head and neck squamous cell carcinoma (HNSCC). Therefore, we have intentionally not included speculations on molecular mechanisms of proteins that differ between the two groups. We hope that the results from this study can contribute to the understanding of the complexity of HNSCC.

No change to the manuscript.

Comment 2. Epidemiological data is needed in the introduction to understand the importance of head and neck cancer.

Response. We agree and have rephrased the first two paragraphs in the Introduction section.

Change to the manuscript, pages 1-2.

Reviewer 2 Report

Dear Authors, 

first of all - congratulations for your great effort and very interesting research, as well as nicely prepared paper. Information given is very consisted, the paper is really well-structured and easy to follow, not too long. I hope my comments will further help you to improve the paper.

TNM paragraph --> it would be nice to mention also the latest viewpoint: genomic medicine approach, oncoagnostic approach. Some people claim that TNM system won't last for long now, with the advent of genomics. The same is true for the IHC testing - while still widely used and relatively cheap, qPCR is much better and sensitive and in many good labs it is being used routinely instead of IHC. It would be nice to discuss this issue more broadly. 

Figure 3 --> would it be possible to have it slightly bigger? It's not easy to analyse it. 

"All but one protein (ARG1) that differed at baseline also displayed lower means after one year. A few earlier studies have assessed the clinical significance of the serum levels of these four proteins in patients with malignant disease." --> Definitely needs clarification and more info, sources etc. Why ARG1 is higher? What is the function of this gene/protein? Any connections with the cancer you're investigating? 

Author Response

Comment 1. TNM paragraph --> it would be nice to mention also the latest viewpoint: genomic medicine approach, oncoagnostic approach. Some people claim that TNM system won't last for long now, with the advent of genomics. The same is true for the IHC testing - while still widely used and relatively cheap, qPCR is much better and sensitive and in many good labs it is being used routinely instead of IHC. It would be nice to discuss this issue more broadly. 

Response. Thank you for this comment. We have added some limited information on more precise outcome prediction. As the goal of the study is to compare the two groups of patients, patients with p16+ and patients with p16- we prefer not to question the use of IHC.

Change to the manuscript, page 2.

Comment 2. Figure 3 --> would it be possible to have it slightly bigger? It's not easy to analyse it. 

Response. To have Figure 3 slightly bigger would negatively impact on the presentation of the findings as the Figure has to be split in two separate Figures. We therefor prefer to keep Figure 3 in the present version.

No change to the manuscript.

Comment 3. All but one protein (ARG1) that differed at baseline also displayed lower means after one year. A few earlier studies have assessed the clinical significance of the serum levels of these four proteins in patients with malignant disease." --> Definitely needs clarification and more info, sources etc. Why ARG1 is higher? What is the function of this gene/protein? Any connections with the cancer you're investigating? 

Response. We agree and have made additional clarifications in the Discussion section on page 7.

Reviewer 3 Report

In the manuscript, the authors identified four differentially expressed proteins between patients with p16+ and p16- tumors. This approach is interesting and, in my opinion, has potential for a diagnostic application for head and neck squamous cell carcinoma. However, some minor improvement of the manuscript is required, especially for Figures.

In the Abstract authors stated, “low expression of four proteins”. The name of these proteins should be mentioned.

The sentence “No proteins showed higher expression in the p16+ group” is not clear. It is related to these five proteins or for five proteins in any tested experiment in the p16+ group.

 Figure 4 should contain data on protein expressions before initiation of treatment and 12 months after treatment.

Some references should support statistical analysis: Fisher’s exact test, the Mann-Whitney test, NPX values, proportional hazards regression, Kaplan–Meier method etc.

Author Response

Comment 1.  In the Abstract authors stated, “low expression of four proteins”. The name of these proteins should be mentioned.

Response. We have added the name of the four proteins as suggested.

Change to the manuscript, page 1.

Comment 2. The sentence “No proteins showed higher expression in the p16+ group” is not clear. It is related to these five proteins or for five proteins in any tested experiment in the p16+ group.

Response: We agree and have rephrased the sentence to make this statement clearer.  

Change to the manuscript, page 2.

Comment 3. Figure 4 should contain data on protein expressions before initiation of treatment and 12 months after treatment.

Regarding baseline data, i.e. statistically significant mean differences in protein expression (NPX) between the 16+ and p16- group before treatment, they are displayed in Figure 1.

No change to the manuscript.